# Human Neurocysticercosis: An Overview

**DOI:** 10.3390/pathogens11101212

**Published:** 2022-10-20

**Authors:** Oscar H. Del Brutto

**Affiliations:** School of Medicine and Research Center, Universidad Espíritu Santo, Urbanización Toscana, Apt 3H, Km 4.5 vía Puntilla, Samborondón 092301, Ecuador; oscardelbrutto@hotmail.com

**Keywords:** neurocysticercosis, cysticercosis, *Taenia solium*, epilepsy, headache, albendazole, praziquantel

## Abstract

Human cysticercosis is caused by ingestion of *T. solium* eggs from taenia carriers. Neurocysticercosis (NCC), defined as the infection of the CNS and the meninges by the larval stage of *Taenia solium*, is the most common helminthic infection of the CNS worldwide. Parasites may lodge in brain parenchyma, subarachnoid space, ventricular system, or spinal cord, causing pathological changes that account for the pleomorphism of this disease. Seizures/epilepsy are the most common clinical manifestation, but other patients present with headache, focal deficits, intracranial hypertension, or cognitive decline. Accurate diagnosis of NCC is possible after interpretation of clinical data together with findings of neuroimaging studies and results of immunological tests. However, neuroimaging studies are fundamental for diagnosis because immunological test and clinical manifestations only provide circumstantial evidence of NCC. The introduction of cysticidal drugs changed the prognosis of most NCC patients. These drugs have been shown to reduce the burden of infection and to improve the clinical course of the disease in many patients. Efforts should be directed to eradicate the disease through the implementation of control programs against all the steps in the life cycle of *T. solium*, including carriers of the adult tapeworm, infected pigs, and eggs in the environment.

## 1. Introduction

Neurocysticercosis (NCC) is defined as the infection of the central nervous system (CNS) and the meninges by the larval stage of *Taenia solium*, the pork tapeworm [1]. NCC is currently widespread and represents the most common helminthic infection of the central nervous system (CNS) [2]. While the prevalence of NCC is not known, it is probable that millions of people are infected by this parasite, and that many of them will experience clinical manifestations of this disease at any point of their lives. NCC is endemic in most Latin American countries, the sub-Saharan Africa, and some regions of Asia. On the contrary, NCC is not common in Northern Europe, the US, Canada, Australia, Japan and New Zealand—except among immigrants—and is eventually reported from Israel and Muslim countries.

Two hosts (human beings and pigs) are involved in the life cycle of *T. solium*. Pigs are an intermediate host and human beings may act as both definitive and intermediate hosts. In most cases, human cysticercosis is the result of the ingestion of *T. solium* eggs directly from taenia carriers (fecal-oral contamination), and eventually due to auto-infection by the fecal-oral route in adults harboring the adult parasite in the intestine. Once in the digestive tract, eggs evolve into oncospheres that are carried through the bloodstream into the CNS and other tissues, where they mature into larval forms or cysticerci. The role of infected pigs is to transmit taeniasis to human beings, but they are not responsible for the occurrence of human cysticercosis [3]. 

Most cases of NCC diagnosed in people living in developed countries are the result of contact with taenia carriers who have migrated from endemic areas [4] or may be seen in international travelers to disease-endemic areas [5]. In these settings, the life cycle of *T. solium* cannot be completed due to adequate husbandry, and human NCC is often observed in isolated cases or in clusters of individuals (often relatives) infected by the same taenia carrier [6]. On the contrary, all conditions favoring the transmission of human taeniasis and the development of human and porcine cysticercosis are found in developing countries (particularly at rural settings). These conditions include: the presence of taenia carriers, inadequate disposal of human feces, free-roaming pigs having access to human feces, and the consumption of undercooked pork. These conditions are often associated with illiteracy, poverty and poor sanitization. Hence, the prevalence of human taeniasis as well as that of human and porcine cysticercosis may reach endemic proportions.

## 2. Clinical Manifestations

Clinical manifestations of NCC depend on the number of lesions, their location within the brain parenchyma, subarachnoid space, ventricular system and spinal cord, as well as the intensity of the immunological reaction against the parasites [7]. Population-based studies conducted in endemic areas have shown that most infected individuals are asymptomatic [8,9], a scenario that differs from that observed in clinical settings, where most NCC patients are evaluated because of seizures/epilepsy (about 70–75% of cases), headache, focal neurological deficits, cognitive decline, or increased intracranial pressure [10].

### 2.1. Seizures/Epilepsy

These manifestations mostly occur in patients with parenchymal NCC, particularly in those with granular or calcified lesions [11]. In patients with granulomas, seizures occur as the result of the breakdown of the blood–brain barrier resulting from inflammation related to the attack of the host’s immune system to previously viable cysts. In patients with calcifications, seizures are probably the result of intermittent liberation of cysticercal antigens trapped within these calcified nodules, which react with antibodies from the host and elicit breakthrough seizures [12]. It has been considered that seizures related to NCC are acute symptomatic. In this view, seizures would disappear when the inflammation subsides. This may occur in some patients with a single colloidal cysticercus located in the brain parenchyma, but only if these lesions quickly disappear [13]. It must be kept in mind that most parasites in the granular and calcified stage represent enduring epileptogenic foci that will cause recurrent unprovoked seizures if the patient is not treated with antiseizure drugs [14].

It is not fully understood why some NCC patients develop seizures/epilepsy while others do not. There are some differences across these categories of patients that may explain the predisposition to develop epilepsy [15]. Individuals with seizures/epilepsy have higher serum levels of matrix metallopeptidase-9, an increased expression of proinflammatory cytokines and lymphocyte adhesion molecules, and an increased probability of mutations in the Toll-like receptor 4 that leads to an increased Th1 (proinflammatory) response compared to those without seizures/epilepsy [16,17,18]. In addition, a more intense serological response to parasite antigens on the enzyme-linked immunoelectrotransfer blot (EITB) assay has been associated with both epilepsy development and with a more severe course once epilepsy has been established [19]. Recent evidence shows other risk factors associated with recurrent (breakthrough) seizures among patients with calcified NCC. These include being female, having calcified lesions in the temporal lobe, having a seizure in the year prior to the first evaluation, having more than 10 lifetime seizure episodes, and having an abnormal interictal EEG (Bustos JA, personal communication).

The actual burden of NCC-related epilepsy is unknown. A population-based study conducted in a rural Ecuadorian village showed that persons with epilepsy had three times the odds of having NCC compared to those without epilepsy [20]. In that population, the crude epilepsy prevalence was 26.8 per 1000 inhabitants, which is higher than that reported from areas non-endemic for cysticercosis [21]. In the same village, the annual incident rate of adult-onset epilepsy was 249 per 100,000 persons-year and individuals with NCC were six times more likely to develop epilepsy compared to those without the disease, with an attributable fraction of incident adult-onset epilepsy due to NCC of 30.9% [22]. These results, together with those of studies coming from other cysticercosis-endemic countries, provide robust evidence that NCC is a major cause of epilepsy in developing countries, accounting for one-third of the excess fraction of epilepsy seen in these regions [23,24].

### 2.2. Headaches

NCC-related headaches have traditionally been associated with intracranial hypertension related to hydrocephalus, subarachnoid or ventricular cysts, or cysticercotic encephalitis [25]. However, these forms of the disease are responsible for the minority of headaches observed in NCC patients. A sizable proportion of subjects with parenchyma brain cysticerci develop headache without evidence of intracranial hypertension [26,27]. Lifetime headache prevalence, current headaches, intense headaches and migrainous headaches, are almost five times more frequent among patients with calcified NCC compared to the matched controls without NCC [28].

Pathogenic mechanisms implicated in the occurrence of migrainous headaches in patients with calcified NCC are not fully understood. Evidence suggest that calcifications contain parasitic membranes that may be presented to the host immune system when structural changes related to calcification remodeling allow antigenic remnants to be in contact with neighboring cerebral tissues [29]. This exposure induces a breakdown in the blood–brain barrier, edema formation and oxidative stress resulting in upregulation of the calcitonin gene-related peptide and other free radicals, which may stimulate the trigemino-vascular reflex, causing a migraine attack [30].

### 2.3. Other Manifestations

NCC may be associated with almost any neurological signs and symptoms. Since NCC is highly pleomorphic, there is no possibility to define a pathognomonic clinical syndrome. Focal neurological deficits have been recognized in about one-third of symptomatic NCC patients [10]. Motor deficits, involuntary movements, Parkinsonism, language disturbances, sensory deficits, and clinical evidence of brainstem dysfunction, may occur in some cases. These manifestations often follow a subacute or chronic course and are most frequently observed in patients with subarachnoid cysts compressing the brain parenchyma [31]. Stroke syndromes also occur in about 3% of NCC patients; these are most often related to cerebral infarctions located in deep cerebral structures [32]. Some patients develop increased intracranial pressure, which is most often related to hydrocephalus secondary to either cysticercotic arachnoiditis, granular ependymitis, or ventricular cysts [31].

Other NCC patients develop cognitive decline, mostly observed in those with associated hippocampal atrophy/sclerosis [33]. Patients with intra or suprasellar cysticerci present with ophthalmologic and endocrinologic disturbances [34]. NCC of the spinal cord is characterized by root pain and weakness when parasites are located in the spinal subarachnoid space, and by motor and sensory deficits for cysts located in the parenchyma of the spinal cord [35]. Subretinal cysticerci is associated with progressive deterioration in visual acuity or visual field defects [36].

## 3. Diagnosis

The diagnosis of NCC is based on neuroimaging findings together with immune diagnostic tests [37]. Clinical manifestations are unspecific and epidemiological data only provide indirect evidence favoring its diagnosis. Identification of the parasite is often not possible due to the variable location of cysticerci within the CNS, and identification of parasites elsewhere in the body should not be taken as a definitive proof of CNS involvement even in subjects with neurological manifestations and suggestive intracranial lesions.

### 3.1. Neuroimaging Studies

Cysticerci have variable appearance on neuroimaging studies [38]. In the brain parenchyma, parasites show different characteristics according to their involutive stage, which include: cystic lesions without enhancement (vesicular cysticerci), cystic or nodular lesions showing abnormal enhancement after contrast medium administration (colloidal and granular cysticerci), and small calcifications (calcified cysticerci). Parasites within the subarachnoid space may appear as cystic lesions that tend to group to each other’s (racemose cysticerci) or may present as a focal or diffuse arachnoiditis that most often involve the Sylvian fissure or the basal CSF cisterns, which are frequently associated with obstructive hydrocephalus. Ventricular cysticerci appear as lesions with different signal properties than the ventricular fluid that distort the anatomy of the ventricular system causing asymmetric hydrocephalus (scolices may be visualized in some of these cysts). Spinal cord cysticerci appear as nodular or cystic lesions if they are located intramedullary or as a focal or diffuse spinal arachnoiditis with or without cystic lesions if located in the spinal subarachnoid space. Figure 1 depicts the most common neuroimaging characteristics of cysticerci within the CNS. Of all these lesions, the only pathognomonic appearance is the presence of vesicular cysts showing the scolex as an eccentric brilliant point (the “hole-with-dot” imaging); however, even in these cases, some cystic tumors may have remnants of neoplastic cells in the interior of the cystic component resembling a scolex (pseudo-scolices). This warning note is of special importance for patients with a single parenchymal brain cyst.

### 3.2. Immune Diagnostic Tests

The enzyme-linked immunosorbent assay (ELISA) for detection of anticysticercal antibodies has been used for NCC diagnosis and is still used in many countries where more sophisticated tests are not available [39]. Recent studies have documented reliability problems with the use of ELISA for anticysticercal antibody detection, particularly due to the use of crude and semi-purified antigenic extracts, and this test should not be considered reliable for cysticercosis diagnosis [40,41].

The electro immunotransfer blot (western blot) assay using lentil lectin purified glycoprotein antigens (LLGP-EITB) is superior to the ELISA for identification of anticysticercal antibodies in serum [42]. This test has a sensitivity of 98% in patients with more than one lesion and does not cross react with antibodies induced by other infections. However, EITB sensitivity drops to about 50% when patients have a single cysticercus, thus limiting the reliability of the test. In addition, the technical complexity of the EITB precludes its worldwide availability [43].

The introduction of monoclonal antibodies-based antigen detection by ELISA is promising for cysticercosis diagnosis. Nevertheless, the diagnostic reliability of antigen detection is limited to patients with viable infections and possible for patients with multiple cysts, predominantly those located in the subarachnoid space or the ventricular system [44]. A recent study suggests that merging information provided by antibody detection by EITB and antigen detection by ELISA may improve the discrimination between viable and non-viable parenchymal brain cysticerci [45]. Likewise, a quantitative polymerase-chain-reaction (qPCR) assay has been developed for NCC diagnosis. This test is useful for confirming the diagnosis of subarachnoid and ventricular NCC [46]. More research is needed to assess the validity of these newly developed tests for the diagnosis of other forms of the disease.

### 3.3. Unification of Diagnostic Criteria

An expert panel developed the first chart of diagnostic criteria for human cysticercosis in 1996. The idea was to elucidate the strength of the different clinical, neuroimaging, and immunological findings observed in patients with cysticercosis, and to outline the varied epidemiological scenarios in which the disease is most likely to occur [47]. This original chart has been revised according to advances in diagnostic tests and the experience achieved with the use of the original chart. The first revision (2001) focused on the diagnosis of NCC because cysticercosis outside the CNS is most often clinically irrelevant [48]. The 2001 version recognized four degrees of diagnostic criteria (absolute, major, minor and epidemiologic) that were stratified according to their diagnostic strength. Proper interpretation of these criteria allowed two levels of diagnostic certainty: definitive and probable [48]. The second revision, published in 2017, modified previous charts on the basis that NCC cannot be diagnosed without direct visualization or histopathological demonstration of the parasite or with the aid of neuroimaging exams [49]. This revised version also included a set of “confirmatory criteria” that can only be established in subsequent neuroimaging evaluations. This set includes: resolution of cysts after the use of cysticidal drugs, spontaneous resolution of single enhancing lesions, and migration of ventricular cysts on sequential neuroimaging studies [49]. This last revision is intended to be used for physicians evaluating patients with NCC at both endemic and non-endemic regions (Table 1). In addition, other complementary exams may be needed in selected cases to exclude alternative pathologies, such as: neurotuberculosis, multiple sclerosis, neurobrucellosis, hydatidosis, brain abscesses, and primary or metastatic brain tumors.

The above-mentioned charts of diagnostic criteria have been accepted by the medical community as proven by their widespread use [37,50]. Some researchers have suggested modifications based on specific diagnostic scenarios, including patients from the Indian subcontinent, patients living in areas where neuroimaging studies are not available, and patients with exclusive extraparenchymal disease [51,52,53]. These adaptations do not offer major advances in NCC diagnosis and may create misunderstandings since the objective of the charts has been the unification of diagnoses that allow comparisons across different study groups [54].

**Table 1 pathogens-11-01212-t001:** Revised Del Brutto’s diagnostic criteria and degrees of diagnostic certainty for neurocysticercosis (*Reproduced from Del Brutto et al., J. Neurol. Sci. 2017, 371; 202–210; Copyright: the authors*).

Diagnostic Criteria
**Absolute Criteria:**
Histological demonstration of the parasite from biopsy of a brain or spinal cord lesion. Visualization of subretinal cysticercus. Conclusive demonstration of a scolex within a cystic lesion on neuroimaging studies.
**Neuroimaging criteria:**
Major neuroimaging criteria: Cystic lesions without a discernible scolex. Enhancing lesions *. Multilobulated cystic lesions in the subarachnoid space. Typical parenchymal brain calcifications *.
Confirmative neuroimaging criteria: Resolution of cystic lesions after cysticidal drug therapy. Spontaneous resolution of single small enhancing lesions ^†^. Migration of ventricular cysts documented on sequential neuroimaging studies *.
Minor neuroimaging criteria: Obstructive hydrocephalus (symmetric or asymmetric) or abnormal enhancement of basal leptomeninges.
**Clinical/exposure criteria:**
Major clinical/exposure: Detection of specific anticysticercal antibodies or cysticercal antigens by well-standardized immunodiagnostic tests *. Cysticercosis outside the central nervous system *. Evidence of a household contact with *T. solium* infection.
Minor clinical/exposure: Clinical manifestations suggestive of neurocysticercosis *. Individuals coming from or living in an area where cysticercosis is endemic *.

**Degrees of diagnostic certainty**
**Definitive Diagnosis:** One absolute criterion.Two major neuroimaging criteria plus any clinical/exposure criteria.One major and one confirmative neuroimaging criteria plus any clinical/exposure criteria.One major neuroimaging criteria plus two clinical/exposure criteria (including at least one major clinical/exposure criterion), together with the exclusion of other pathologies producing similar neuroimaging findings.
**Probable Diagnosis:** One major neuroimaging criteria plus any two clinical/exposure criteriaOne minor neuroimaging criteria plus at least one major clinical/exposure criteria.

*** Operational Definitions.** *Cystic lesions:* rounded, well-defined lesions with liquid contents of signal similar to that of CSF on CT or MRI; *Enhancing lesions:* single or multiple, ring- or nodular-enhancing lesions of 10–20 mm in diameter, with or without surrounding edema, but not displacing midline structures; *Typical parenchymal brain calcifications:* single or multiple, solid, and most usually <10 mm in diameter; *Migration of ventricular cyst:* Demonstration of a different location of ventricular cystic lesions on sequential CTs or MRIs; *Well-standardized immunodiagnostic tests:* so far, antibody detection by enzyme-linked immunoelectrotransfer blot assay using lentil lectin-purified *T. solium* antigens, and detection of cysticercal antigens by monoclonal antibody-based ELISA; *Cysticercosis outside the central nervous system:* demonstration of cysticerci from biopsy of subcutaneous nodules, X-ray films or CT showing cigar-shape calcifications in soft tissues, or visualization of the parasite in the anterior chamber of the eye; *Suggestive clinical manifestations:* mainly seizures (often starting in individuals aged 20–49 years; the diagnosis of seizures in this context is not excluded if patients are outside of the typical age range), but other manifestations include chronic headaches, focal neurologic deficits, intracranial hypertension and cognitive decline; *Cysticercosis-endemic area:* a place where active transmission is documented.

^†^ The use of corticosteroids makes this criterion invalid.

## 4. Treatment

In view of the pleomorphism of NCC, a unified therapeutic approach is not useful in all patients. A rational approach must be tailored according to the viability and location of cysticerci, and the severity of the host’s immune response to the parasites [50,55,56]. The first line of management must always be directed to the control of clinical manifestations (seizures, headache, and intracranial hypertension) and pathogenic mechanisms involved in their occurrence (brain edema, inflammation, compressive effects or hydrocephalus). This approach includes the use of antiseizure medications, anti-inflammatory drugs, corticosteroids, corticosteroid-sparing agents and etanercept, alone or in combination [57,58,59,60]. Surgical procedures, such as cysts’ resection, ventricular shunt placements, and decompressive craniotomies, are still needed in some cases [61,62,63].

Praziquantel and albendazole have improved the prognosis of many patients with NCC. However, the anecdotal nature of the initial trials with these cysticidal drugs generated controversies [64]. More recently, well-designed trials have shown that the use of albendazole and praziquantel results in the resolution of most viable cysticerci located in the brain parenchyma and also improve clinical manifestations in these patients [65,66,67]. Cysticidals should not be used in patients with parenchymal brain calcifications and are contraindicated in patients with cysticercotic encephalitis or in those presenting with the “starry-sky” appearance of parenchymal NCC. In these cases, their use may cause deleterious reactions due to the enhancement of the host inflammatory reaction against parasites [68].

Treatment of extraparenchymal NCC is complicated since many patients do not respond to traditional doses of cysticidal drugs. In these cases, increased dosages, prolonged administration, or repeated drug trials may be needed to destroy the cysts (Figure 2). However, no randomized controlled trials confirming the benefits and risks of therapy for extraparenchymal NCC have been published, and all the available evidence is based on expert opinion and non-controlled trials [31,62,63,69].

Recent guidelines from an expert group provided comprehensive evidence of the different therapeutic approaches for the diverse forms of NCC [70]. With the exception of therapy for patients with viable parenchymal brain cysticerci, most of the knowledge is based on non-controlled studies or small case series (Table 2). These guidelines recommended the simultaneous use of albendazole and praziquantel for patients with more than two viable cysts in the brain parenchyma, based on the evidence provided by two double-blind controlled trials [71,72]. However, one of those trials showed that the difference in the efficacy of albendazole plus praziquantel is not significantly better than the use of high doses of albendazole alone (22.2 mg/kg/day instead of 15 mg/kg/day) [71]. The latter should be used in countries where praziquantel is not available.

The rationale for using combined therapy is based on distinct mechanism of actions of the two cysticidal drugs [56]. Praziquantel affects calcium channels in the parasite’s surface and produces muscle contractions, paralysis and tegumental damage. Albendazole interferes with cell division of the parasite, leads to degeneration of parasite cytoplasmic microtubules, affects ATP formation, and impairs glucose intake leading to energy depletion.

## 5. Control Measures

Neurocysticercosis is endemic in regions where conditions favoring the transmission of *T. solium* are found. As previously noticed, these include poor disposal of human feces, illiteracy, slaughtering of pigs without veterinary control, and the presence of free-roaming pigs [73]. Cysticercosis is a potentially eradicable disease. However, eradication programs must be directed to all the interconnected steps in disease transmission, including taenia carriers, infected pigs, and eggs in the environment [74,75]. Inadequate coverage of one of these steps may result in a rebound in the prevalence of taeniasis/cysticercosis after the program has been completed.

Intervention programs for cysticercosis elimination often target NCC-related epilepsy as a potentially eradicable condition [76,77]. Even if these programs succeed in interrupting NCC transmission, they do not represent the solution on the problem of NCC-related epilepsy. Residual NCC-related epilepsy in patients with parenchymal brain calcified cysticerci may persist for the lifespan of the patient. An example of this is the situation reported from a rural Ecuadorian village where cysticercosis transmission has long been arrested [78]. In this setting, where almost 10% of the population aged ≥20 years have calcified NCC (and no living cysts have been noticed on neuroimaging studies), the actual incidence risk ratio of epilepsy is higher than 200 per 100,000 persons-year, and the attributable fraction of incident adult-onset epilepsy due to NCC is higher than 30% [22].

Two other aspects of control should be mentioned in this review. One is the development of vaccines for the prevention of disease in pigs, in particular one using a oncospheres antigen (TSOL18), which has been shown to be highly effective to prevent cysticercosis infection [78]. The other is the recently introduced CystiHuman, a model aiming to get informed decision-making through cost–benefit analyses about different interventions to NCC control [79].

## 6. Final Remarks

While our knowledge on NCC has improved over the past two decades, recent advances do not represent the final word on diagnostic and therapeutic approaches to this parasitic disease. There are some questions that will probably be fixed over the next years. The last revision of diagnostic criteria for NCC proved reliable for the diagnosis of ventricular NCC [80], but this version must be validated for other forms of the disease. The introduction of an immune diagnostic test with 100% sensitivity and specificity for all forms of the disease is far to be accomplished. This will be particularly helpful in remote rural settings where sophisticated technology for NCC diagnosis is not available. Likewise, some doubts remain about the optimal length of therapy with antiseizure medications for patients with NCC-related epilepsy. The efficacy of cysticidal drugs for patients with subarachnoid and ventricular NCC has not been evaluated by means of well-conducted randomized controlled trials [81]. Finally, more research is needed for a better understanding of long-term sequelae of calcified cysticerci in the brain parenchyma, which may be associated with breakthrough seizures, recurrent migrainous attacks or with the development of secondary hippocampal atrophy/sclerosis [29,82,83].

## Figures and Tables

**Figure 1 pathogens-11-01212-f001:**
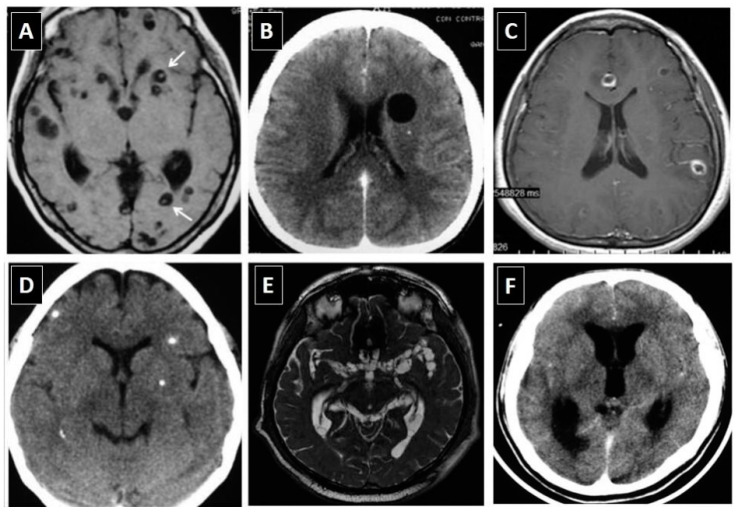
Neuroimaging characteristics of neurocysticercosis: (**A**) Parenchymal vesicular cysts showing the scolex (arrows) in T1-weighted MRI; (**B**) Vesicular cyst without scolex in contrast-enhanced CT scan; (**C**) Colloidal cysticerci appearing as ring-enhancing lesions in gadolinium-enhanced T1-weighted MRI; (**D**) Parenchymal brain calcifications in non-enhanced CT scan; (**E**) Multilobulated (racemose) subarachnoid cysticerci in FIESTA sequence MRI; and (**F**) Abnormal enhancement of basal leptomeninges and hydrocephalus in contrast-enhanced CT scan (*Reproduced from Del Brutto et al, J Neurol Sci 2017, 371, 202-210) Copyright: the authors*).

**Figure 2 pathogens-11-01212-f002:**
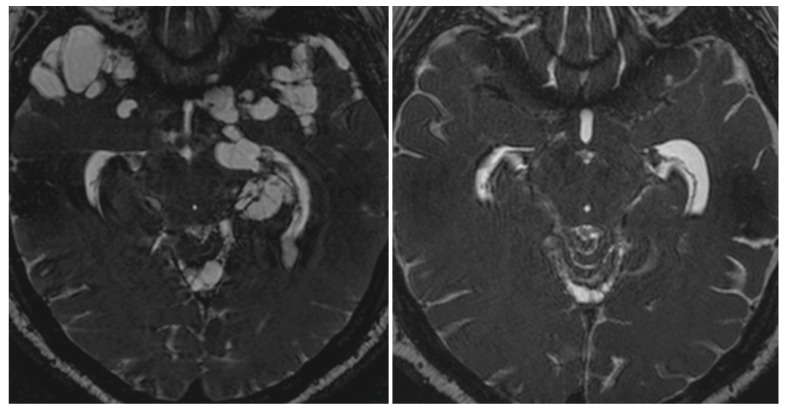
Fast Imaging Employing Steady-state Acquisition (FIESTA) MRI sequence showing massive cysticercal infection of the base of the brain and temporal pole by subarachnoid cysticerci (**left**) and their disappearance six months after cysticidal drug therapy (**right**). Courtesy of the Cysticercosis Working Group in Perú.

**Table 2 pathogens-11-01212-t002:** General guidelines for therapy of neurocysticercosis. Level 1 of evidence favors the use of cysticidal drugs in patients with parenchymal brain vesicular and colloidal cysts. For other forms of the disease, guidelines are based on Levels 2 and 3 of evidence.

**PARENCHYMAL NEUROCYSTICERCOSIS** **Vesicular cysts:** **One or two cysts:** Albendazole 15 mg/kg/day (up to 1200 mg/day) for 10 days. Corticosteroids often needed. AED for seizures. **More than two cysts:** Albendazole 15 mg/kg/day (up to 1200 mg/day) plus praziquantel (50 mg/kg/day) for 10 to 14 days. Albendazole alone at higher doses (22.2 mg/kg/day) for 10 days may be used when praziquantel is not available. Corticosteroids must be used. AED for seizures. **Heavy infections:** Albendazole 15 mg/kg/day (up to 1200 mg/day) plus praziquantel (50 mg/kg/day) for 10 to 14 days (repeated cycles may be needed). Corticosteroids are mandatory before, during, and after therapy. AED for seizures. **Colloidal cysts:** **Single cyst:** Albendazole 15 mg/kg/day (up to 800 mg/day) for 10–14 days. Corticosteroids may be used when necessary. AED for seizures. **Mild to moderate infections:** 15 mg/kg/day (up to 800 mg/day) for 10–14 days. Praziquantel may be added (50 mg/kg/day for 10–14 days). Corticosteroids are usually needed before and during therapy. AED for seizures. **Cysticercotic encephalitis:** Cysticidal drugs are contraindicated during the acute phase of the disease. Corticosteroids and osmotic diuretics indicated to reduce brain swelling. AED for seizures. Decompressive craniectomy in refractory cases. **Granular and calcified cysticerci** **Single or multiple:** No need for cysticidal drug therapy. AED for seizures. Corticosteroids for patients with recurrent seizures and perilesional edema surrounding calcifications.
** EXTRAPARENCHYMAL NEUROCYSTICERCOSIS ** **Small cysts over convexity of cerebral hemispheres:** **Single or multiple:** Albendazole 15 mg/kg/day (up to 1200 mg/day) for 10–14 days for one or two cysts. Praziquantel (50 mg/kg/day) for 10 to 14 days may be added when more than two cysts. Corticosteroids may be used when necessary. AED for seizures. **Cysts in Sylvian fissures or basal CSF cisterns:** **Racemose cysticercus:** Albendazole, 20 to 30 mg/kg/day for 15 to 30 days (repeated cycles of albendazole may be needed). Praziquantel (50 mg/kg/day) may be added. Corticosteroids are mandatory before, during, and after therapy. **Other forms of extraparenchymal neurocysticercosis:** **Hydrocephalus:** No need for cysticidal drugs therapy. Ventricular shunt. Continuous corticosteroid administration (50 mg prednisone three times a week for up to two years) may be needed to reduce the rate of shunt dysfunction. **Ventricular cysts:** Endoscopic resection of cysts in the lateral or third ventricles. Suboccipital craniectomy for fourth ventricle cysts. Albendazole may be used only in small lesions located in lateral ventricles when cysts are attached to the ventricular wall. Ventricular shunt needed in patients with associated ependymitis. **Angiitis, chronic arachnoiditis:** No need for cysticidal drug therapy. Corticosteroids are mandatory. **Cysticercosis of the spinal subarachnoid space:** Albendazole, 20 to 30 mg/kg/day for 15 to 30 days (repeated cycles of albendazole may be needed). Praziquantel (50 mg/kg/day) may be added. Corticosteroids are mandatory before, during, and after therapy. Surgical resection of lesions may be considered. **Intramedullary cysticercosis:** Albendazole 15 mg/kg/day (up to 1200 mg/day) for 10 days. Corticosteroids always needed. **Intrasellar cysticerci:** Surgical resection of lesions (anecdotal evidence). No evidence of albendazole efficacy. **Subretinal cysticerci:** Surgical resection of lesions (anecdotal evidence).

## Data Availability

Data presented here is supported by references.

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
