# Peer review of "Human Neurocysticercosis: An Overview"

_pathogens, 2022, doi:10.3390/pathogens11101212_

Round 1

Reviewer 1 Report

I suggest to add some information on the global epidemiology of NCC

Introduction--> line 5: sate the intermediate and definitive host (pigs and humans)

Clinical manifestations---> line 5: a scenario

Author Response

Thank you for your comments. I agree with all of them.

1.- information on global epidemiology has been added in the introduction section.

2.- the role of pigs and humans as intermediated and definitive hosts have been clarified.

3.- the typo was corrected (a scenario)

Reviewer 2 Report

This is a well-written article by an expert in cysticercosis with extensive experience, including a valuable update on the diagnostic criteria for neurocysticercosis.

However, certain information on some aspects of the disease should be improved. Specifically, I encourage the author to include the following in his overview:

a.     More detailed information on the life cycle of the parasite.

b.     Information on all the different modes of transmission of human cysticercosis, including the possibility of autoinfection in Taenia solium carriers.

c.     Data on the burden of the disease worldwide, not only in endemic areas but also in new ones due to the phenomenon of globalization.

d.     Information on pig vaccines and pilot control strategies in the control measure section.

e.     Brief information on CystyHuman, the first model of human neurocysticercosis.

Author Response

Thank you very much for your useful comments. I agree with all of them.

We added information on all the requested points. Some of them were already requested by the other reviewers. Please, see the corrected proofs. 

Reviewer 3 Report

This review was written for a special issue on T solium, Cysticercosis. It is comprehensive overview on human NCC providing an updated summary of different aspects such as clinical presentation, diagnosis and treatment. Even if many similar reviews on this topic have been already published (the majority by the same author that represent one of the main experts on this topic!) this review is really didactic and very useful for a correct clinical approach.

I have no major concerns just few suggestions:

-              Figure one should be better described detailing one by one all the six pictures included and adding an arrow to display the lesions

-              In the "Final remark" section", probably it should be underlined that unfortunately in rural areas of endemic countries current diagnostic criteria are still scarcely applicable, due to the lack of neuroimaging and serological tests, underlining the need of a quick and cheap test with adequate sensitivity and specificity.

Author Response

Thank you for your comment. I agree with all of them.

1.- legend for figure 1 was not included in the manuscript that you reviewed. I have modified this in the revised submission. Arrows cannot be added since this figure has been reproduced from a previous paper (permission was granted as mentioned in the legend of the figure).

2.- I added the requested comment in the final remarks section